# String Tension based Borewell Water Level Monitoring using IoT

Tanmay Bhatt, Krishna Singh, Kaustubh Singh

**Abstract**

The high rate of groundwater usage has resulted in a rapid decline in groundwater levels, which has necessitated its monitoring. This project focuses on estimating the groundwater level inside borewells remotely and in real time using the Internet of Things (IoT). For this, a solution is proposed which is based on string tension and does not require any electrical components to be lowered into the borewell. The solution provides a completely automated, real-time, reliable, and IoT-enabled alternative to existing methods for borewell water monitoring. The proposed approach is compared with an ultrasonic sensor-based approach in a controlled environment as well as in a tank. Additionally, four nodes were deployed in a small educational campus in the Indian city of Hyderabad to ascertain its usability and reliability in practical situations. The observations from the data collected over a month show that the proposed low-cost solution is reliable and has a good performance in the field.

## 1  Introduction

With freshwater sources like rivers and lakes being localized to a certain geographical location, groundwater becomes an indispensable source of freshwater. In this context, wells and borewells play a very important role in fulfilling the day-to-day water requirement of potable water. Borewells are basically vertically drilled wells which are bored into an underground aquifer to extract water for various purposes. Nowadays, groundwater scarcity has become a very pressing issue and needs to be solved imperatively [1]. Monitoring the water level in borewells gives us valuable information about the aquifers and the groundwater.

The challenge in measuring the water level in the borewell lies in the dimensions of the borewell. Most borewells have a small diameter of about 30-40 cm in comparison to the depth of the well, which is in the range of a few metres to hundreds of metres (5-150 m). As old borewells dry up, newer and deeper borewells are dug, which sometimes even reach up to a depth of 400 m. This makes it very difficult to measure the level inside said borewells.

Even after the tremendous advancement of technology, some manual methods for water level estimation still exist, which are still very prevalent, especially in countries like India. These include dropping a stone and calculating the time taken for it to hit the water, lowering down a string tied to a stone and then measuring the length of the dry string [2], and lowering two electrical wires connected to a buzzer in the borewell and measuring the length of the wire when the buzzer makes a sound. Although easy to use and simple, these manual methods provide little reliability due to the chance of manual error and require someone to go to the borewell location and perform the measurement each time. This makes real-time monitoring infeasible.

Here's where IoT devices can come to the rescue. With an increase in the popularity of IoT, these devices which integrate these day-to-day tasks with the internet have become commonplace. They provide us with a means to be involved with the surroundings at a level which was previously unheard of [3]. These devices decrease manual workload and generate valuable data, allowing users to make smarter and more aware decisions [4]. Along with these benefits, IoT devices can also be controlled and monitored using smart devices like mobiles and tablets remotely, which makes them very flexible and convenient to use [5].

There have been few commercially available solutions which use sensors and IoT methods for estimating borewell operations [6, 7, 8, 9]. In [6], a method is presented to allow users to control their submersible pumps via the internet and alert them when the pump is not reaching the water level. [7] uses the motor's temperature to determine whether the motor is actually pumping water or running dry, preventing the motor from burning out. It also provides information about the pH and turbidity of the water. [8] is an app-based approach which relies on geolocation data to estimate water levels inside the borewell. [9] is a wired

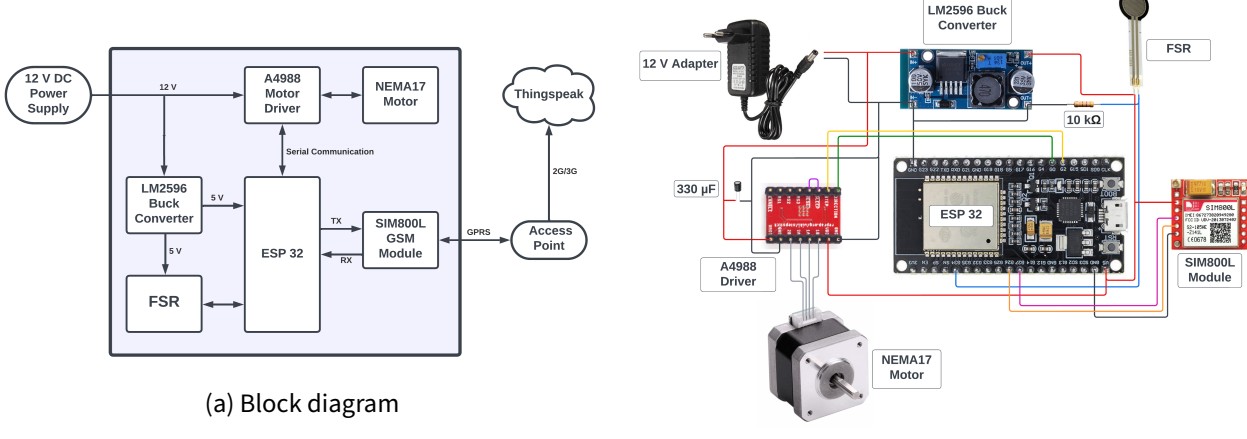

(a) Block diagram

(b) Circuit diagram

Figure 1: Block and circuit diagram of device node

solution which uses a piezoelectric sensor which is lowered into the borewell, and the pressure data is used to calculate the water level inside the borewell.

Note that [6, 7] do not actually measure the water depth. The solutions in [8, 9] do not have IoT capability; hence, someone still needs to go to the location and note down the readings. Moreover, since the sensor in [9] is wired, procuring the required length of wire becomes expensive and troublesome for deeper borewells.

Unlike [6, 7, 8, 9], this project focuses on estimating borewell water level using a completely automated, non-wired, real-time, IoT-enabled solution.

The specific contributions of this project are

- An IoT-based device is developed which relies on string tension to estimate the depth at which water is present in a borewell. The proposed solution can also be used for detecting water levels in traditional wells as well as sumps and overhead tanks.

- The project also presents practical design considerations such as maintaining pre-requisite tension in the string, recalibration mechanism and power consumption.

- The performance of the developed device is compared with an ultrasonic sensor in a controlled environment (known distances) as well as in an overhead water tank to showcase the reliability and superior performance of the proposed device.

- Four nodes were deployed on the borewells of our campus for over a month, collecting more than $10,000$ data points for each node, which establishes the device's usability in practical situations.

## 2   Goals

The major goal of this project was to implement an IoT-based solution which can give us accurate measurements of the water level in a borewell whilst avoiding the use of manual labour. In the following section, we will expand on these points.

Through the use of our string mechanism IoT device, we've been able to solve these issues and make it easy to collect data at a low cost. The device can take readings at regular intervals and push them to the cloud. It does not require us to dip any electric component in the water. The device also recalibrates after a fixed time to maintain the accuracy of the readings. Therefore the device qualifies all the important metrics required for a good IoT device, which includes automation, accuracy, connectivity to the cloud, low power consumption and cost-effectiveness

Table 1: Force sensor specifications

| Parameter | Lower Limit | Upper Limit |
|---|---|---|
| Resistance | 200 Ω | Infinite |
| Force | 0 N | 100 N |
| Current Usage | 0 A | 1 mA |

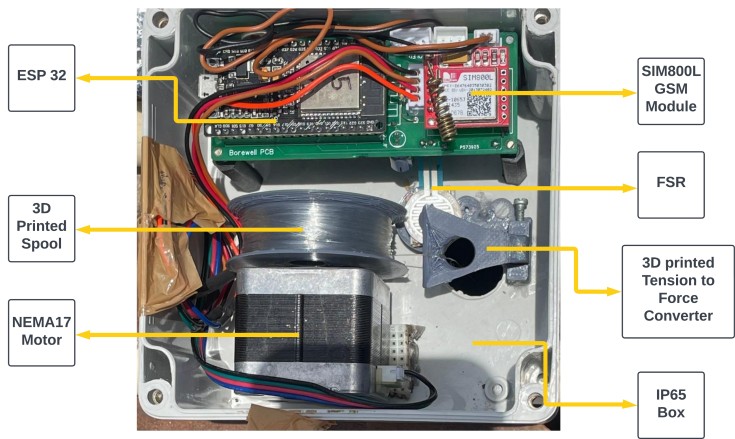

Figure 2: Assembled node in IP65 box

Figure 3: Deployment inside the IIITH campus.

## 3 Implementation and Deployment

### 3.1 Sensor Node Implementation

Fig. 1 shows the block diagram and the circuit diagram of the proposed device. As can be observed in Fig. 1(a), each node consists of a force-sensitive resistor (FSR), a NEMA17 stepper motor and A4988 stepper motor driver, a SIM800L GSM Module and an LM2596 DC-DC buck converter. A Wi-Fi-enabled ESP32 microcontroller is used to interface these components. The sensor used to measure the tension in the string here is a force-sensitive resistor (FSR) (see Table 1 for specification [10]). The resistance of the FSR varies in proportion to the amount of force being exerted on it. The FSR value needs to be converted into a voltage value, which the microcontroller can read. This is done by using a potential divider circuit with a known resistor (10 kΩ in our case). The sensor communicates with the microcontroller via SPI, which in turn communicates with the GSM module via SPI. The SIM800L GSM Module establishes the GPRS connection and connects to the cellular tower which then pushes the data to the *Thingspeak* server [11], which is a cloud-based IoT platform for storing, processing and displaying data, via 2G/3G.

The device is powered with the help of a 12 V DC adapter. As shown in Fig. 1(b), The 12 V output is directly provided to the NEMA17 stepper motor through the A4988 stepper motor driver. Moreover, the 12 V is stepped down to 5 V with the help of an LM2596 DC-DC Buck converter. This 5 V output is used to power the ESP32 microcontroller and the SIM800L GSM Module and to build the potential divider used for force sensing. This apparatus is then enclosed inside an IP65 box, making it possible to deploy the nodes in the field by providing resistance against weather conditions.

The optimal sensing interval was found to be 1 hour after performing frequency analysis on the collected data and finding the Nyquist frequency of the changes in water level, but it is important to note that for the purpose of this project, the sensing interval has been kept as 5 minutes so that a larger amount of data can be captured while simultaneously testing wear and tear of the device.

In order to facilitate the measurement of string tension with the help of the FSR, we have designed and 3D printed a tension-force conversion contraption and spool, which can be seen in Fig. 2. The string passes through the hole in the middle, and when it is under tension, the top piece presses on the platform where the FSR is placed. This causes a force to be applied on the FSR, which can then be measured. Hence, we can

convert the tension in the string into a force which the FSR can measure.

Along with this, we have also designed a spool, as seen in Fig. 2, to dispense the string using the stepper motor. The tapered design of the spool makes sure that the string does not unravel on its own and cause problems. Moreover, since the dimensions of the spool are known, we can calculate the length of the string, which is wound up or dispensed in a single rotation.

## 3.2    Deployment in the field

The fully assembled node is shown in Fig. 2. Four such nodes were created and deployed in various locations in the campus of IIIT-H, Gachibowli, Hyderabad, India, the locations of which are indicated in Fig. 3. Deploying the devices in different locations allows us to gather data about the water level in different places as well as check the reliability of the devices themselves. Node-4 (Vindhya), which is shown in Fig. 3, was deployed in an overhead tank along with an ultrasonic sensor-based water level meter in order to compare and contrast the readings of both devices in an environment where the water level changes in a much smaller time frame as compared to borewell water levels. Fig. 4 shows pictures of the actual deployed devices. Node-1 has been deployed for a period of four months, since April 2023, and has collected around $25,000$ data points in order to observe seasonal data, while Node-2 and Node-3 were deployed by June 2023 and have collected around $8,000$ data points after successful data collection by Node-1. Node-4 was deployed along with the ultrasonic sensor-based meter for a one-week duration in order to compare the reliability of the readings over the course of around $1,000$ data points. It is essential to note here that all the nodes were deployed in borewells where there is no motor installed to pump the water out. This means that the water level here changes only due to environmental factors.

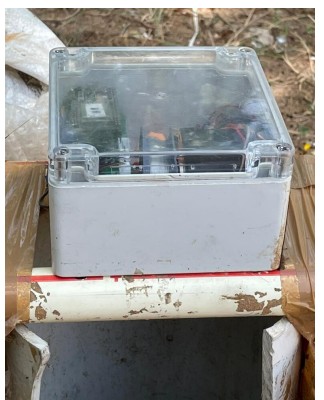

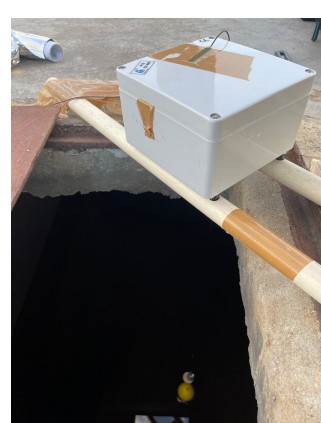

(a) Deployed in Borewell                              (b) Deployed in Tank

Figure 4: Deployment in field

## 4    Working Principle

The main idea behind the working is the presence or absence of tension in a string connected to a bob as can be seen in Fig. 5. We build a potential divider which consists of the FSR acting as the pull-up resistor connected to 5 V and the known resistor, which is 10 k$\Omega$ in this case, acting as the pull-down resistor connected to the ground. The value of this resistor sets the sensitivity of the force sensor; hence, based on the bob's weight, we can set this value. When pressure is applied on the FSR, its resistance decreases, meaning the voltage at the potential divider output will increase. This output is connected to an analog read pin of the microcontroller, thus allowing us to read the changes in the voltage value. Using this arrangement, we can quantify the pressure being applied on the FSR as changes in the voltage level at the output of the potential divider. The bob is lowered down into the borewell with the help of the stepper motor as shown in Fig. 5. The bob's weight causes the string to have some non-zero amount of tension which is reflected by a non-zero value returned by the FSR. When the bob reaches the water's surface, it starts floating on the surface. At this

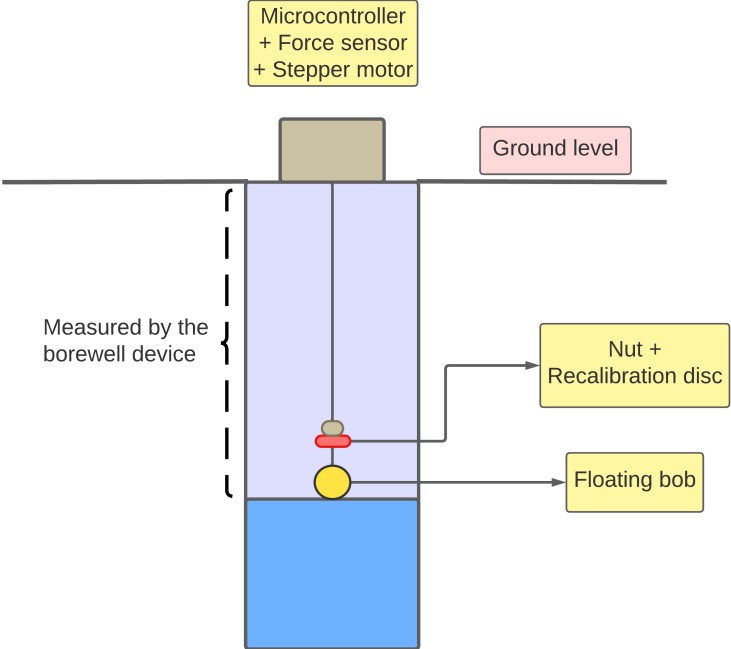

Figure 5: Schematic Diagram

point, since the bob's weight is counteracted by buoyant force, the tension in the string becomes zero. This is reflected by a very small value returned by the FSR, hence allowing us to detect the water level.

An important thing to notice here is that the proposed device measures the length of the string from the ground level. This means that an increase in the reading means that the water level has gone down, whereas a decrease in the reading means that the water level has risen up. This is shown via the schematic diagram in Fig. 5.

## 4.1 Calculation of the water level

In order to calculate the actual water level, three things are needed: initial position of the bob, circumference of the spool and number of steps rotated by the spool via the motor.

Assuming that initially the distance is set to 0 and hence the motor starts unwinding the string, and the length of string unwound can be calculated using the formulas given below. In our case, the spool has a radius of $r = 20$ mm and the motor takes 200 steps for one full rotation. This means that for each step, the length of string wound/unwound in 1 step is given by

$$l = \frac{C}{200} = \frac{2\pi r}{200} = 0.628 \text{ mm}, \tag{1}$$

where $C = 2\pi r$ is the circumference of the spool. Assuming that initially the distance is set to 0, the motor moves down $N$ steps, then the water level $L$ (relative to the device position on the top) can be given by the length of the string which will be unwound as

$$L = Nl. \tag{2}$$

## 4.2 Adpating to changes in water level

The main sensing element of the device is a floating bob suspended by a nylon string inside the borewell. Now, we can only sense the tension in the string with the help of the FSR; based on that, we need to figure out where the bob is with respect to the actual water level. There can be three different situations in this case which are discussed in the following subsections.

### 4.2.1  Water level has gone down, and the bob is hanging above the water surface

In this situation, the water level has decreased, and hence the bob is hanging above the water level when the microcontroller wakes up after the 5-minute sensing interval. At this point, the bob exerts weight on the string, and hence there is some tension in the string. This means that the FSR will return a non-zero value. The microcontroller senses this and signals the stepper motor to unwind the string. This manoeuvre will stop when the bob reaches the water level and starts floating. At this point, the tension in the string will become zero, and hence the FSR will return 0. The microcontroller will signal the motor to stop unwinding the string. Since we knew the previous water level as well as the length of the string, which was unwound, we can calculate the new water level.

$$L_{N+1} = L_N + Nl. \tag{3}$$

### 4.2.2  Water level has increased, and the bob is floating on the surface

In this situation, the water level has gone up, and hence the bob is floating at a position higher than it was during the previous reading. When the microcontroller wakes up, the bob will be floating on the surface of the water, and hence there will be no tension in the string. This means that the FSR will return a zero value. At this point, the microcontroller will signal the motor to start winding the string. As soon as the bob leaves the water surface, there will be tension in the string, and the FSR reading will become some non-zero value. This means that the bob is now in a hanging position, which is the same as in the previous case. The microcontroller now tells the motor to start unwinding the string till the FSR reading becomes zero. Now, since the water level has gone up, the length of the string wound back will be greater than the length of the unwound string. Thus, we get a level which is smaller than the previous reading as

$$L_{N+1} = L_N - Nl. \tag{4}$$

### 4.2.3  Water level has not changed, and the bob is floating on the surface

In this situation, the water level has remained the same since the last reading. The procedure followed is the same as the one in the previous case where the water level had increased. The only difference lies in the fact that the water level has not changed, and hence the length of the string that is unwound will be the same as the length of the string that was wound initially, thus keeping the water level reading the same such that

$$L_{N+1} = L_N. \tag{5}$$

## 5  Data Pre-Processing

After the data is sent to the Thingspeak platform, it needs to be processed in order to analyze it further. The processing can be split into two major parts, removal of outliers and filling data gaps, which is discussed in the subsequent subsections.

## 5.1  Removal of outliers and smoothing

The water level data sent by the devices will inevitably have some outliers which need to be taken care of to perform a proper analysis. This is done by first thresholding the data around the mean value with the threshold placed at $3\sigma$ ($\sigma$ is the variance of the data) to remove extreme outliers which might have occurred due to some malfunctioning in the device. The next step is traversing the data in windows and calculating the median of the window which gets rid of any accidental outliers. After this is done, a moving average filter with an interval size of 20 for Nodes 1, 2 and 3 and an interval size of 8 for Node-4 is applied over the data

while which smoothens the curve and gets rid of sharp edges in the water depth data. The window sizes were decided based on the nature of the data via trial and error method.

## 5.2    Filling gaps in the data

Sometimes due to power failures or other malfunctions in the device, data loss happens, and gaps are generated in the data. These gaps need to be properly taken care of during analysis so that a reasonable picture of the changes in water level can be created. In order to do this, we have filled the gaps in the data with the local average of the surrounding readings. This ensures that the readings in the gap zone are more or less of the same order as the surrounding readings, which is reasonable since the borewell water level depth does not change abruptly.

# 6    Practical Considerations

This section briefly describes the considerations to be kept in mind while actually deploying the device in a practical environment.

## 6.1    Maintaining pre-requisite tension in the string

While winding the string up and back onto the spool, there needs to be at least some amount of tension in the string otherwise, due to the slack, the string might miss the spool and get muddled up in the motor shaft causing the whole system to fail. When the bob is floating on the surface of the water, there is no tension in the string and hence the problem described above might occur.

In order to get around this, a small weight in the form of a stainless steel nut has been put on the string just above the bob as shown in Fig. 5. This ensures that even when the bob is floating on the water surface, the metal nut still hangs from the string to maintain enough tension to wind the string up without any issues. The weight of the nut is not so much so as to affect the reading of the FSR to an extent which interferes with the regular functioning of the device.

## 6.2    Re-calibration

In case of a power cut or a manual reset, the previous reading of the water level is lost, and hence it is not possible to follow the same steps discussed in the previous section since all of them require the prior reading to calculate the current one. To solve this issue, there is a re-calibration mechanism implemented in the device. Whenever there is a complete restart, be it due to some power cut or a manual reset, the microcontroller will signal the motor to start winding up the string. This happens until the bob reaches the very top of the borewell where the device is placed. There is a disc attached to the string just above the bob which is shown in Fig. 5. The diameter of this disc is slightly larger than the diameter of the hole drilled into the IP65 box to allow the string to pass through. This means that the disc will not be able to enter the box and will be obstructed midway. Now the motor is trying to wind the string up but the disc is causing an obstruction. This obstruction causes the tension in the string to further increase to a value which is greater than that when just the weight of the hanging bob is acting on it. This difference in the tension level is reflected by the FSR reading. The microcontroller interprets this high FSR value as a re-calibration signal and sets the water level value to 0. From here on, the normal operation of the device can continue. Thus, this procedure ensures that the device gets properly re-calibrated in case of a reset.

## 6.3    Power consumption

The power consumed by the device in different states is summarized in Table 2. It can be seen that a considerable amount of power is required even when the device is idle in order to keep the stepper motor fixed in

Table 2: Power Consumption

| Process | Current Consumed | Power | Duration) (per hour) | Energy Consumed |
|---|---|---|---|---|
| Motor is Idle | 0.41 A | 4.92 W | 58 min | 4.75 Wh |
| Motor is Rotating | 0.33 A | 3.96 W | 2 min | 0.13 Wh |
| Data Upload | 0.29 A | 3.48 W | 30 sec | 0.03 Wh |

its current position. This makes using a battery as a power source infeasible. Hence, the device needs to be powered using a 12 V DC adapter which is connected to the electrical mains. Since Borewells already need a power supply to power the submersible pump, electrical outlets are readily available at the site.

## 6.4   Cost considerations

Along with being reliable and convenient, the device requires it to be affordable to be adopted in the mainstream market, especially in Indian markets. The proposed borewell device can be manufactured for about ₹4000−₹5000 ($50 − $60), which is much cheaper as compared to existing solutions in the market.

## 7   Performance Evaluation and Testing Results

This subsection presents some results collected on the field by the deployed nodes in the IIIT Hyderabad campus. Fig. 8 shows the data collected by node one regarding the changes in water level, along with precipitation data collected over a period of three months. It can be seen that a spike in the relative intensity of rain is followed by a dip in the water depth. This means that the rainwater is charging the underground aquifer, which is reflected by the change in the water level. Similarly, Fig. 9 depicts the water level data collected by node two over a smaller time frame of a month, with light rainfall, which can be seen by the smaller changes in water level. Once again, the dips characterize precipitation in the surrounding areas. The third node was deployed in a dry borewell which had been cut off from its aquifer, and hence it just gave a constant reading. The resolution of the readings was kept to 5 cm during these experiments but can be tuned according to the application and requirement.

## 8   Availability

Codes - https://iiitaphyd-my.sharepoint.com/:u:/g/personal/tanmay_b_research_iiit_ac_in/ EUT-OgMUfBpFgys_hWCeIPoBDEl_vXCRDO_R-feyCVnpug?e=KFHPa2

Presentation Video - https://iiitaphyd-my.sharepoint.com/:v:/g/personal/tanmay_b_research_ iiit_ac_in/ERa83Xw7dgZJvT7G9aXnbjsBJtZRy_qwIy5x2oZj18WG9g?nav=eyJyZWZlcnJhbEluZm8iOnsicmVmZXJyYW e=nV5KIP

## 9   Concluding Remarks and Avenues for Future Work

This project proposes a smart, IoT-enabled solution for real-time water level monitoring in borewells which is based on string tension. The same device can also be used to measure the water level in tanks, wells, or any water source which has an open water surface. The proposed solution was found to give much higher precision as compared to existing solutions in the market and can be tuned according to the application as well. The reliability was also tested with the help of an ultrasonic water meter in a controlled environment, where it was shown that an ultrasonic sensor-based meter is not usable for enclosed spaces like borewells due to the reverberation of sound or for deeper borewells where ultrasonic measurements become very inaccurate. The proposed solution provides a low-cost, IoT-enabled, automatic water-level monitoring alternative while keeping practical considerations and overall feasibility in mind.

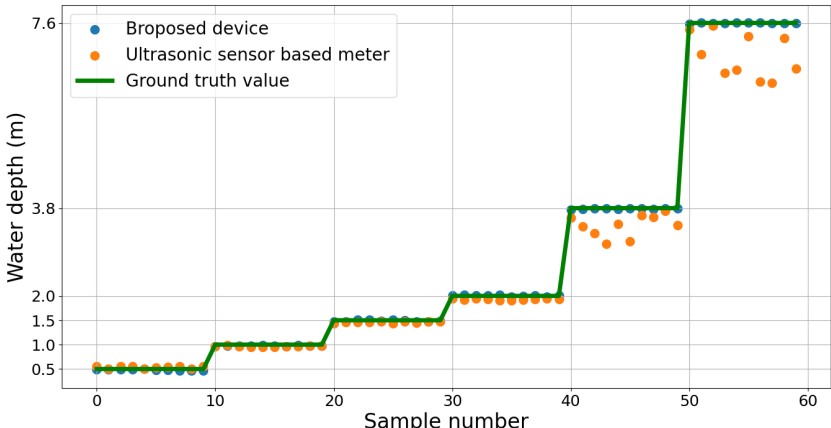

Figure 6: Comparison of borewell device and ultrasonic sensor based meter in a controlled environment

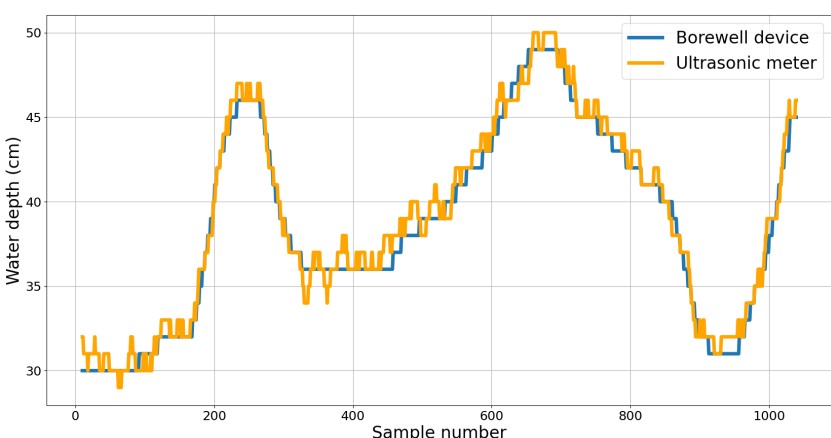

Figure 7: Comparison of the borewell device and ultrasonic sensor based meter in a tank

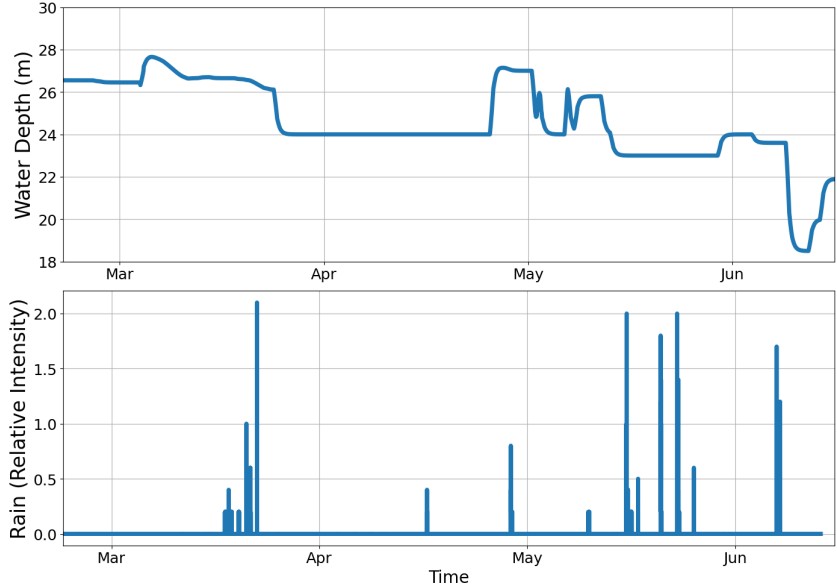

Figure 8: Node 1 water depth vs precipitation data.

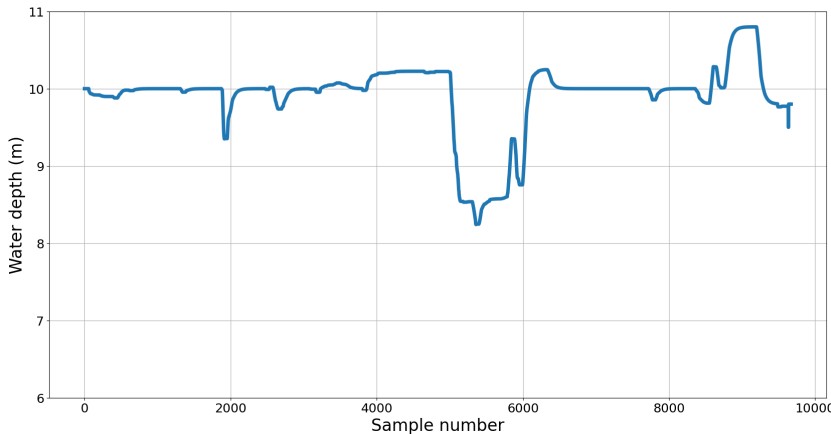

Figure 9: Node 2 water depth data

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
