# OpenReview forum: "String Tension based Borewell Water Level Monitoring using IoT"
_helsinki.fi/ESPC/2023/Competition — ESPC 2023 LongPresentation_

### Official Review · Reviewer_ERMq · 2023-11-09

**Rating:** 3
**Confidence:** 3

**Summary:**

The authors present their approach to measure the water level using sensors with minimal human intervention.
Monitoring the ground water levels is important given the increasing dependence on ground water as a source of fresh water.
The authors have deployed their solution in wells and have analyzed the data collected over a period of 4 months.
It would be nice to see follow up works where the authors further improve the efficiency of the system, and in particular reduce the power consumption when the system is idle.

**Strengths:**

- The work is timely given the increasing usage of ground water. It also improves the current state of the art practices which are largely imprecise and require manual intervention. The need for an automated solution and the challenges given the dimension of the wells is detailed in the introduction.

- The hardware architecture is detailed in Figure 1 and the schematic diagram that summarizes the working is presented in Figure 5. The figures are easy to understand.
-The system has been deployed in wells for a period of 4 months. This longitudinal deployment hints at the robustness of the work.
- The report is clear and the video does a good job in demonstrating the the solution in a concise manner (see nitpicks for issues with the video).
-  The evaluation is thorough. The system performance is evaluated against the ground truth, and it is also compared with other state-of-the-art approaches.

**Weaknesses:**

- There is limited discussion on the software architecture of the system. It would have been nice to see pointers to the configurations etc and the thinkspeak messages formats and intervals used in the source code repository.
- It is not clear if the microcontroller needs to support WiFi. The usage of Wi-Fi  (if used an an alternative to the GSM) is not clear.

Nitpicks
- It would have been nice if the video was not sped up to meet the 10 minute deadline. The speeding up made the content difficult to understand what the presenter is saying.
- The URLs in the availability section do not point to the repository and video.
- The source code only points to the plotting scripts.
- It would have been nice to see a discussion on the reasons for the outliers, and the motivation for the choice of windows for the moving averages. What were the factors that affected the choice of the window and the filters.

---

### Official Review · Reviewer_Bdai · 2023-11-17

**Rating:** 3
**Confidence:** 3

**Summary:**

In this work, a prototype for monitoring borewell water level is developed with the help of IoT.
The developed IoT node consists of a force-sensitive resistor, a stepper motor, GSM module, DC-DC buck converter, and ESP32 with WiFi. The collected data is pushed to ThingSpeak server for further analysis.

The is a bob that is sent to the borewell with a string with the help of a stepper motor. When it reaches the water surface, the tension  in the string becomes zero due to water's buoyancy that is captured by force sensitive resistor that helps authors to measure the water level.

**Strengths:**

*Prototype is working and is tested on multiple sites on their university campus
*Authors have considered the practical challenges of cost and battery usage
*Data is collected and sent to cloud for further processing

**Weaknesses:**

*Figure 6 - has a spelling mistake in the label - Broposed* and The terms used in Figure 6 and Figure 7 are different for the same device
*How is ground truth obtained in the plots shown?
*No discussion on how accurate the results are

---

### Official Review · Reviewer_YyPN · 2023-11-20

**Rating:** 4
**Confidence:** 3

**Summary:**

This project estimated the groundwater level inside borewells remotely and in real time using the Internet of Things (IoT). The solution is  based on string tension and does not require any electrical components to be lowered into the borewell. The prototype appears to be  automated, real-time, reliable, and IoT-enabled alternative to existing methods for borewell water monitoring. The proposed approach was compared with an ultrasonic sensor- based approach in a controlled environment.

**Strengths:**

This is a strong project with the right components required for research.  The introduction is well written supported by references, and the contributions to research are outlines. The goal of creating an end-to-end IoT solution is clear. The experiment design - the deployment description is replicable. Three scenarios are described and it is easy follow in the paper and the accompanied video. I am not sure if the practical considerations are part of the results or discussions. The performance testing the remarks for the future are interesting although this section can be reordered. The video was well produced, although the voice was hurried. For future, I suggest less words. It is good to see the software has been shared.

**Weaknesses:**

This is interesting project. It would good ton further understand the general need for measuring the borehole water depth. The problems of the existing methods used. The results and analysis, and  discussion and conclusion sections are messy- The information needs ordered better. The discussion could have explored the sensitivity of the prototype to problem, such as if the string gets a little tangle (caught) or wet (damaged).